# Prediction of Railway Track Condition for Preventive Maintenance by Using a Data-Driven Approach

Cecília Vale *[ID] and Maria Lurdes Simões [ID]

CONSTRUCT, Faculty of Engineering (FEUP), University of Porto, Rua Dr. Roberto Frias,
4200-456 Porto, Portugal; lurdes.simoes@fe.up.pt
* Correspondence: cvale@fe.up.pt

**Abstract:** For scheduling track maintenance, infrastructure managers perform inspections to assess the track condition. When the irregularities are higher than the threshold limits, a track has a defect that should be corrected to avoid future failures or traffic disruption. Scheduling maintenance actions contribute to reliability and availability but demand the prediction of the evolution of track degradation. In recent years, several degradation models have been defined to forecast geometrical evolution over time and/or tonnage, mainly for heavy rail systems. Nevertheless, most of those models have limitations when dealing with measured data collected in different time intervals as happens in reality. To overcome this problem, a data-driven model based on the logistic binary function is presented and validated with real inspection measurements. The results prove that the model has a 91.1% success rate, an excellent discrimination ability, and a high sensitivity, classifying correctly 84.1% of inspections in need of maintenance. The model also has high specificity as it classifies 94.5% of inspections with no demand of maintenance action. The model is easy to implement, which is also an advantage for the track asset management with guaranty of excellent sensitivity and discrimination.

**Keywords:** track-degradation model; scheduling maintenance; data-driven approach; logistic binary model; principal component analysis





## 1. Introduction

A reliable railway track is vital for safety, train punctuality, passenger comfort, and cost-effectiveness of maintenance and renewal activities, and for that, railway asset management is needed. This management involves all railway systems and components and comprises the most suitable methods, procedures, and tools to optimize costs, performance, and risks for the rail infrastructure life cycle. In operation, railway tracks are subjected to traffic loads that lead to railway track geometrical degradation. When the track irregularities are higher than the legal threshold limits, this means that the track presents a defect that can cause failure or traffic disruption whose consequences can be significant, including a high cost of railway maintenance, economic loss during operation, damage to the railway asset or on the rail vehicle, and accidents with possible loss of human lives. Hence, the prediction of degradation is a key phase for the definition of inspection and maintenance plans. Track-condition monitoring is crucial to preserving the performance of railway infrastructure assets [1]. Figure 1 shows that, when a geometrical track indicator is reaching the legal threshold, a maintenance action needs to be performed. These actions are performed to reduce or eliminate possible failures and to restore a failed railway part to an operational state.

To schedule maintenance actions, the actual track condition, as well as expected evolution of that condition, should be known through inspection data. However, inspection and maintenance plans deal with various uncertainties. To overcome the randomness and the uncertainties of the track-degradation phenomena, probabilistic models are usually

considered. As the track condition advances over time, a stochastic methodology is often used. Furthermore, Markov models are often adopted for the prediction of track degradation as a probabilistic method. However, although this methodology is a very common probabilistic approach to predict the track condition, it has two main limitations: (i) the basic assumption of the Markov process is that the probability of going from a condition to any other condition depends only on the current state condition and not on the history of track condition, which is as relevant in some cases as the railway track degradation; (ii) Markov models are based on system condition defined at constant time intervals, which is not also a realistic situation in the railway field, since track inspections are not usually conducted at constant intervals.

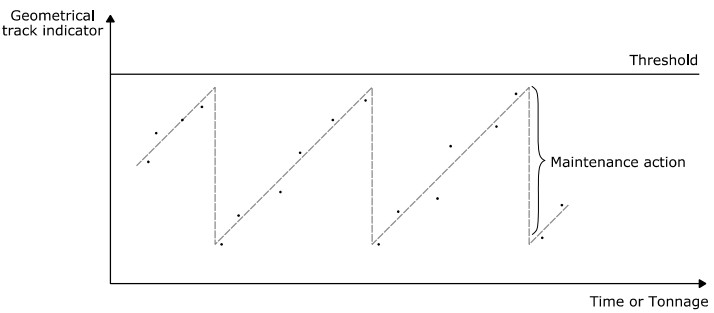

**Figure 1.** Track degradation and maintenance.

To overcome these two major limitations, in this research a new insight is put forward in an attempt to predict the condition of a railway track from a point of view of distributed defects for forthcoming condition-based maintenance application based on the history of track condition defined at any time, i.e., with no constant inspection time interval.

This paper is organized as follows. Section 2 focuses on the geometrical track quality, by presenting the track-condition assessment according to the European Standard EN 13848-5 [2] and by reviewing geometrical tack degradation models. Section 3 describes the defined methodology for predicting the track geometrical condition. Section 4 shows an application of the methodology to a real railway track with validation. Finally, Section 5 presents the conclusions.

## 2. Geometrical Track Quality

### 2.1. Condition Assessment

Track geometry is characterized by five geometrical parameters: longitudinal level, alignment, gauge, cross level, and twist. Figure 2a–e show the definitions of the deviations of those five parameters.

The assessment of geometrical track quality can be conducted by analyzing these five geometrical indicators separately as proposed by the European Standard EN 13848-5 [2], or by track quality indexes ($TQI_s$) that might combine the track irregularities in two or more dimensions. There are several $TQI_s$ used around the world and a summarized overview of them is presented below.

For the Federal Railroad Administration (FRA), a $TQI$ is calculated for gauge, alignment, longitudinal level, and cross level [3], according to the equation below:

$$TQI_s = \left( \frac{L_s}{L_0} - 1 \right) \times 10^6 \qquad (1)$$

where $L_0$ is the theoretical length of the track section and $L_s$ is the traced length of the space curve calculated by Equation (2), with $\Delta_x$ being the sample space and $\Delta_y$ the difference between two consecutive measurements.

$$L_s = \sum_{i=1}^{n} \sqrt{\Delta_{xi}^2 + \Delta_{yi}^2} \qquad (2)$$

This *TQI* does not combine any of the track geometrical indicators, which means that it considers a single parameter. However, some other railway administrations use $TQI_s$ that combine two or more geometrical parameters into a single measure of ride quality.

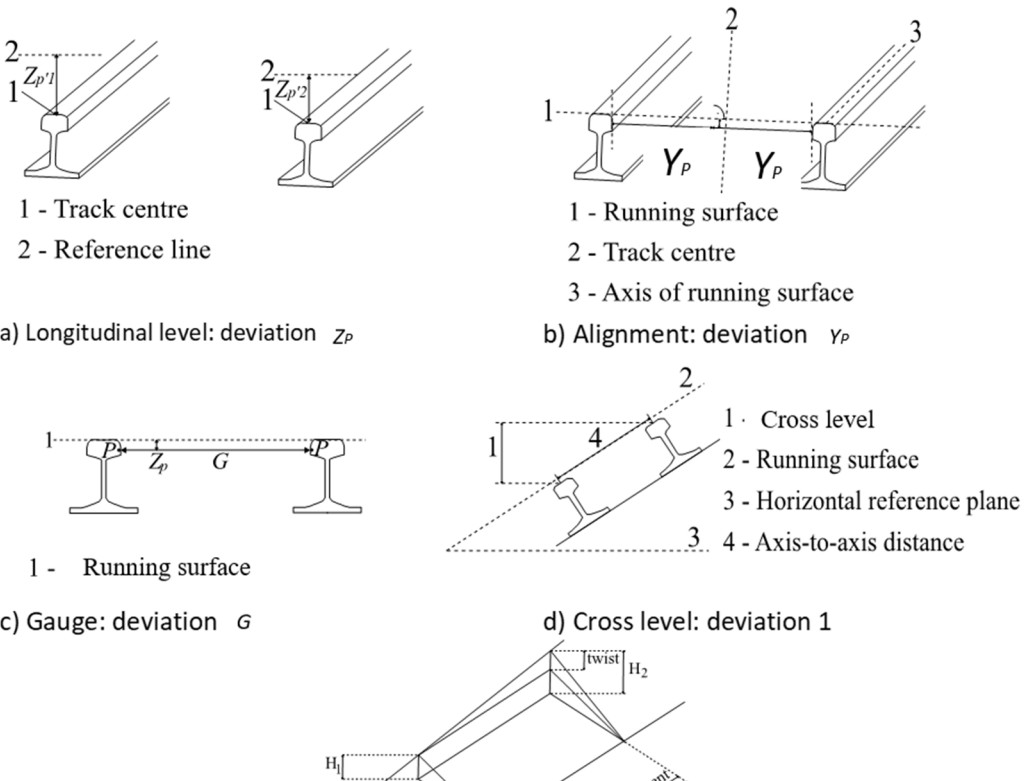

Figure 2. Track geometry parameters: (**a**) Longitudinal level; (**b**) Alignment; (**c**) Gauge; (**d**) Cross level; (**e**) Twist.

The J-synthetic coefficient [4] is a track quality indicator developed by the Polish railways and it can be calculated as presented in Equation (3).

$$\frac{S_z + S_y + S_w + 0.5 S_e}{3.5} \tag{3}$$

where $S_z$, $S_y$, $S_w$, and $S_e$ are the standard deviation of the vertical irregularities, horizontal irregularities, twist, and gauge, respectively.

Besides these TQIs, Offenbacher et al. [5] presented fourteen track indicators used by railway infrastructure managers in different countries and applied them for a real five-kilometer test section.

More recently, Mahsa and Mohammadzadeh [6] propose a stochastic track quality index that considers the uncertainty regarding the quality classification. For that, the authors consider a Bayesian framework associated with a Monte Carlo simulation which is applied to 900 km of railway tracks.

In Europe, track quality assessment should be performed according to the European Standard EN 13848-5 [2], which compares the deviations of longitudinal level, alignment, gauge, cross level, and twist with defined thresholds. The standard EN 13848-5 [2] states that geometrical track quality assessment should be analyzed not only in terms of

distributed irregularities, but also in terms of isolated defects being the type of defect characterized by its wavelength, amplitude, and shape.

In Figure 3, a synthesis of the geometrical track quality assessment according to the European Standard EN13848-5 [2] is presented.

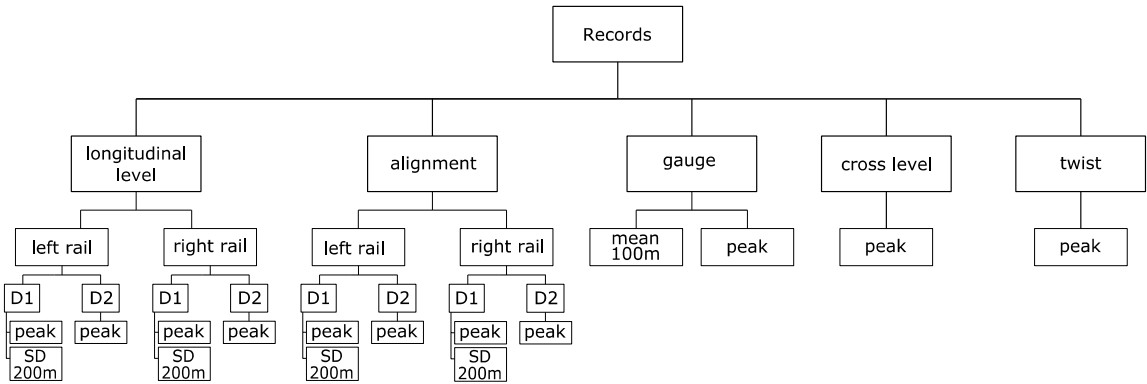

**Figure 3.** Geometrical track quality assessment.

The measured data, which consist of records of longitudinal level, alignment in both rails, gauge, cross level, and twist, are collected by inspection vehicle. The longitudinal level and the alignment records are filtered in two wavelength ($\lambda$) ranges: D1 with $3 \leq \lambda \leq 25$ m and D2 with $25 \leq \lambda \leq 70$ m. With the filtered track profile, some quality indicators should be calculated depending on the type of defect in analysis: distributed or isolated defect (Figures 4 and 5). In these two Figures, NL represents the longitudinal level and x the track position.

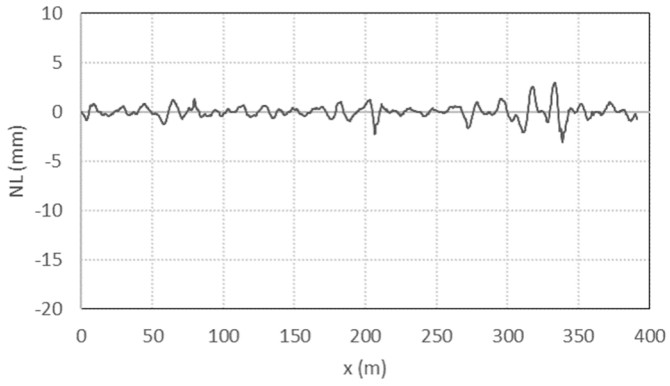

**Figure 4.** Distributed irregularity of the longitudinal level.

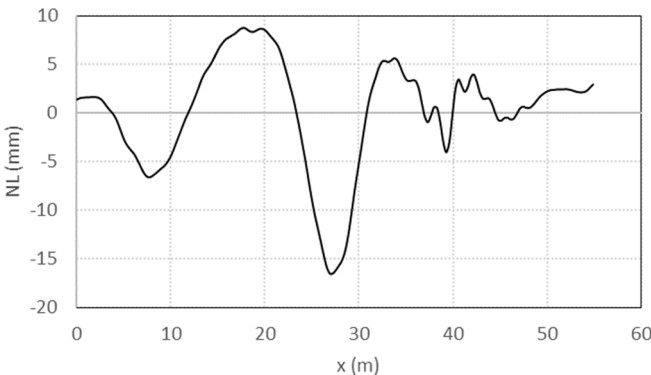

**Figure 5.** Isolated irregularity of the longitudinal level.

The indicators for geometrical track quality assessment according to the European standard are:

i.  the standard deviation (SD) of the longitudinal level of left and right rails over track segments of 200 m;
ii.  the standard deviation (SD) of the alignment of left and right rails over track segments of 200 m;
iii.  the mean gauge over track segments of 100 m;
iv.  the peak amplitude of longitudinal level defects with wavelength between 3 and 25 m (D1);
v.  the peak amplitude of longitudinal level defects with wavelength between 25 and 70 m (D2);
vi.  the peak amplitude of alignment defects with wavelength between 3 and 25 m (D1);
vii.  the peak amplitude of alignment defects with wavelength between 25 and 70 m (D2);
viii.  the peak gauge value;
ix.  the peak cross level value;
x.  the peak twist value.

The track indicators mentioned in i., ii., and iii. characterize the distributed irregularities over a track segment of 200 m or 100 m, while the other indicators are defined to characterize isolated (or peak) defects of the tracks such as dipped weld joint.

To say that an indicator corresponds to a defect, limits should be defined. For that purpose, EN13848-5 [2] refers to three thresholds given as a function of speed:

(1)  safety limit, given only for isolated defects, and if an irregularity exceeds this limit, it is necessary to take immediate measures, such as lowering the maximum speed of trains or closing the line until the defect has been corrected;
(2)  intervention limit, given only for isolated defects, and if an irregularity exceeds this limit, it is necessary to conduct corrective maintenance actions so that the safety limit is not reached before the next inspection;
(3)  alert limit, given for both distributed and isolated defects, and if an irregularity exceeds these, regular planned maintenance operations need to be scheduled.

### 2.2. A Review of Rail Track-Degradation Models

In recent years, several degradation models have been defined in order to predict geometrical track evolution over time and tonnage, mainly for heavy rail systems.

Several authors [7–9] provide a comprehensive review of rail-degradation prediction models and classify degradation models as mechanistic models, statistic models, mechanical–empiric models and artificial-intelligence models. Mechanistic models are based on the knowledge and understanding of the behavior of the mechanical components. In terms of mechanistic and mechanical–empiric models, the most-known models are the Shenton [10,11], the Sato [12,13], and the ORE model [14]. However, these mechanical–empiric models have some limitations: (1) the majority of them only consider vertical degradation of the track, disregarding lateral degradation; (2) these models do not consider the dynamic interaction between the train and the track, which contributes significantly to track geometrical degradation, as posited by Vale and Calçada [15].

Statistical models are based on data collected from track inspections, and they can be categorized into deterministic, probabilistic, or stochastic models. In statistical models, the variables used as inputs are usually traffic (speed, tonnage, volume), axle loads, rail type, and maintenance data. These models work well for large datasets, but they do not account for degradation uncertainty. An application of a deterministic model can be found in [16] for the optimization of preventive maintenance actions in ballasted tracks. Probabilistic, models by dealing with large numbers of datasets, are able to achieve a more accurate degradation evolution; however, to achieve accuracy, a large number of historical data are needed. As far as the stochastic models are concerned, these have been defined in recent years, but focusing exclusively on distributed track irregularities characterized by a single condition indicator. Vale and Simões [17] defined a stochastic model based on the Dagum

distribution to characterize the geometrical track-degradation process over time of distributed irregularities of longitudinal level. Later, Vale and Ribeiro [18] applied this model to a condition-based maintenance model formulated as a mixed 0–1 nonlinear program.

For alignment irregularities, Kawaguchi et al. [19] proposed two degradation models. The first is based on lateral deformation of the track to estimate the time to maintenance, and the second model predicts track alignment irregularities for a time frame of one year by using the exponential smoothing method.

Track quality indexes ($TQI_s$) that combine track irregularities in two or more dimensions [4] were the object of study by Lasisi and Attoh-Okine [20], who applied principal component analysis to obtain a small set of variables that allows the prediction of defects and reveals other features in the track geometry data in addition to the combined $TQI$, although there were some correlations potentially useful for track maintenance.

More recently, Letot et al. [21] built a degradation model that uses a random coefficient Wiener degradation-based process and considers a probabilistic model to simulate the recovery effect after the maintenance action for defining adaptive maintenance scheduling based on track-condition prediction.

Stochastic Markov modeling has also been used by several authors [22–24]. However, this approach has a significant limitation when dealing with measured data collected in different time intervals as happens in real situations. Furthermore, in Markov models, the actual track condition is defined based only on the previous track condition and the probability of transition between condition states, which means that the history of track condition is disregarded.

In some cases, the deterioration phenomena depend on several causes and exhibit different fault modes. To consider these aspects, Bian et al. [25] propose a fault-diagnosis method based on a self-organizing feature-map network and support-vector machine, focusing on the use of non-fault data to identify degradation states under different fault modes. This approach considers four phases: data acquisition, feature processing, state mining, and state identification, and it is defined for specific railway points such as turnouts. In other research, Jia and Gardoni [26] propose a renewal-theory life-cycle analysis with state-dependent stochastic models that describe the deterioration processes. Although this approach has been applied and validated for a reinforced concrete bridge subject to deterioration due to corrosion and seismic loading, it has potential for railway tracks as these systems also have degradation depending on several interactions such as traffic, tonnage, weather conditions, or seismic damage, and are subjected to renewal activities to restore track performance.

Degradation modeling of track geometry is important in designing an optimization-based maintenance schedule. Vale et al. [16,18] applied, for the optimal maintenance scheduling of a Portuguese railway track, deterministic and stochastic degradation models. More recently, Bressi et al. [24] applied a stochastic degradation model to an optimization procedure for an Italian railway track. Sharma et al. [27] developed a data-driven, condition-based policy for the inspection and maintenance of track geometry by using TQIs. Gerum et al. [28] present an approach for defect prediction in railways when faulty data are present or detailed data are not available, and integrate that prediction with the scheduling of rail inspections and maintenances.

Another type of degradation modeling is artificial intelligence models: artificial neural networks (ANNs) and neuro-fuzzy, which are machine learning models [29], are not described because they are out of scope of this research.

## 3. Methodology for Predicting Track Geometrical Condition

In this section, a data-driven approach is used to predict railway track condition based on the history of track condition and on inspection measurements obtained in non-constant time intervals. This is an innovated probabilistic methodology aiming to model track degradation by considering the particular aspects of track inspections as realistically as possible.

The data-driven methodology is presented in Figure 6 and comprises two phases. The goal of the training phase is to create an accurate model that correctly represents the evolution of the track degradation. For this phase, we need to collect data. Prior to the prediction, the data-driven model needs to be trained on historical data to capture the statistical relationships between independent variables (input) and the dependent variables (output). During the prediction phase, the railway track condition is predicted for the next inspection time(s) based on a set of historical data.

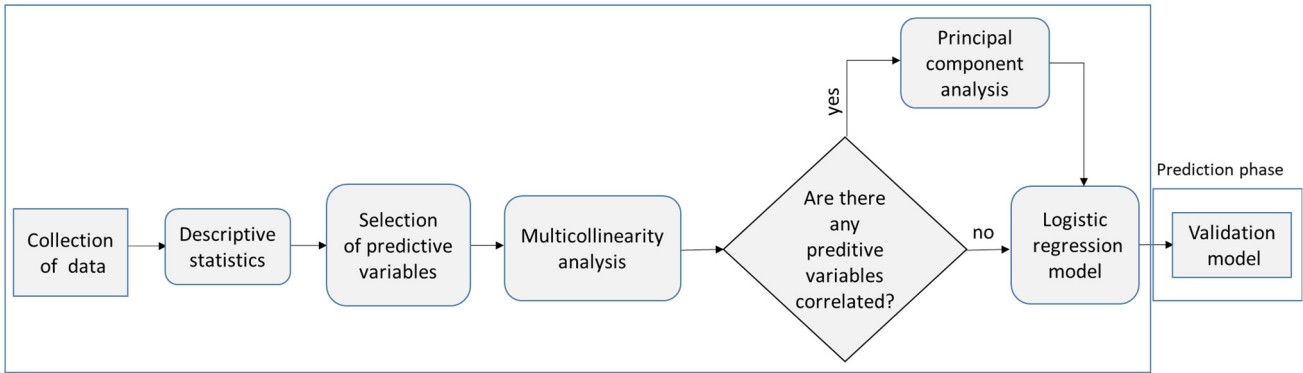

**Figure 6.** Block diagram of the data-driven approach to predict railway track condition.

To forecast railway track condition, represented by *C*, the logistic function was chosen. Logistic binary regression expresses the probability of occurrence of a defined track quality condition by taking values 0 (good track) or 1 (track with defect). This probabilistic model also performs better than the linear regression, which is not appropriate for modeling variables that are not normally distributed. Considering the logistic binary regression, the probability distribution of a particular condition can be calculated by the logit function expressed by Equation (4).

$$P(C) = \frac{e^{\alpha + \beta_1 X_1 + \ldots + \beta_n X_n}}{1 + e^{\alpha + \beta_1 X_1 + \ldots + \beta_n X_n}} \tag{4}$$

where *P(C)* is the probability of occurrence of condition *C*, $\alpha$ is an intercept-related function constant, $\beta_i$ are slope-related function constants, and $X_i$ (*i = 1, ... , n*) are the independent variables of the function.

In the present study, binary logistic models were used to predict condition (dependent variable) because, for deciding if a maintenance action is needed or not, it is sufficient to know whether a track segment condition is below a defined limit (*C = 0*) or not (*C = 1*). The potential independent variables (predictors) considered in the model are the standard deviation of the longitudinal level of the right and left rail (respectively, $SDLL_R$ and $SDLL_L$) at each inspection time, the standard deviation of the alignment of the right and left rail at each inspection time (respectively, $SDA_R$ and $SDA_L$), the time interval, in days, between the inspection time and the previous time inspection (NDAYS), and the sequential number of track segments for modeling (TSEG). These predictors were selected based on the European standard EN 13848-5 [2].

Prior to the logistic regression modeling, the assumptions were validated by performing the diagnosis of collinearity between independent variables. This diagnosis includes analysis of the linear correlation (*r*) and the calculus of the variance inflation factors (VIF). VIF values higher than 10 indicate that the variables are highly correlated [30]. The statistical significance of the correlation between the dependent and the independent variables was tested by using a significance level of 5%. The resulting *p*-values (>5%) indicate that the correlation have no statistical significance and because of that the independent variable was removed. To solve the problem of collinearity between the predictive variables without

removing some of them of the model, principal component analysis (PCA) was applied. This analysis was used to reduce the dataset and to define the representative variables to model. This exploratory data technique enables not only elimination of the existing collinearity between the key variables, but also representation of the left and right measures, for the longitudinal level and the alignment, by a single variable.

To evaluate the quality of the fitted models, the $r^2$ coefficients of Cox & Snell and Nagelkerke were used, as well as the sensitivity and specificity. The binary logistic regression was performed by using IBM SPSS statistics software, version 25. With the SPSS statistical software, it is also possible to compare the observed values with the predicted values by fitted models. The resulting comparisons give the total percentage of prediction success of the model. The modeling is performed with data inspections of the N−2 instant times and the validation of the obtained model are made using the last two data inspections.

This methodology is easy to apply and enables a simple model that characterizes the quality or degradation of the tracks in future moments with high confidence. The application of principal component analysis makes it possible to overcome the obstacle of collinearity between key parameters in the characterization of the track condition without eliminating the variability inherent to each one of these parameters.

## 4. Application

### 4.1. Data

The geometrical track-condition parameters (longitudinal level and alignment) are measured by an inspection vehicle that periodically checks the quality of the railway tracks. In a first phase, these measurements are raw data associated to the track position through GPS. As there is always a small error in GPS coordination of different inspections, when combining raw data from different campaigns a synchronization of the records is needed. As track curvature does not change over time it is used as the base to perform the synchronization of records. In Figure 7a, the curvature profiles of a railway track in several campaigns are shown before synchronization. As it can be seen, there are some offsets between the records that should not exist, since the track curvature is always the same for a given track. Therefore, considering that the track curvature does not change with time, data synchronization can be completed by applying the cross-correlation function to the curvature of each campaign measurement. Figure 7b shows the track curvature after performing synchronization, and now all the records are superposed.

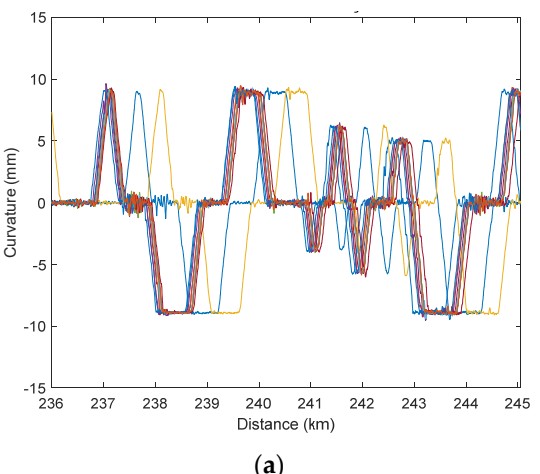

(**a**)

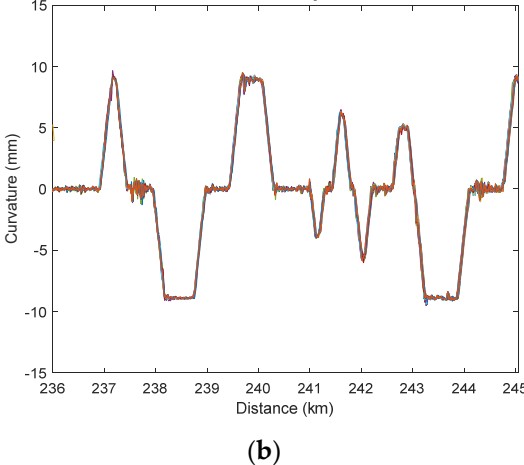

(**b**)

**Figure 7.** Curvature before (**a**) and after (**b**) synchronization.

After identifying the offset of each inspection campaign in relation to a base campaign, the identified value can be applied to all of the other condition indicators (longitudinal level, alignment, etc.) of that specific inspection in order to obtain the synchronized measurements of all the geometrical indicators. In Figure 8, the synchronization phase for a track segment of 9 km is presented for the longitudinal level of the right rail of the railway track in study, as an example.

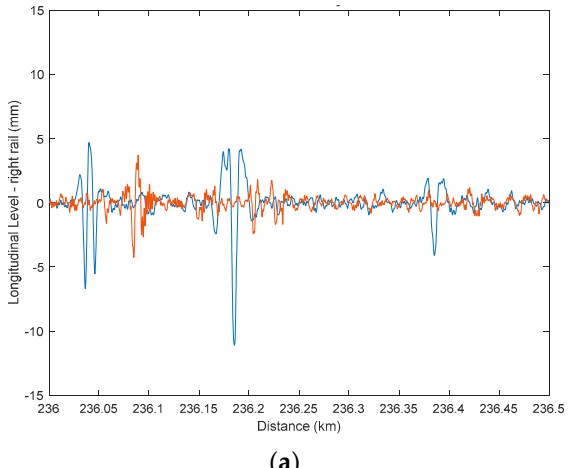

(**a**)

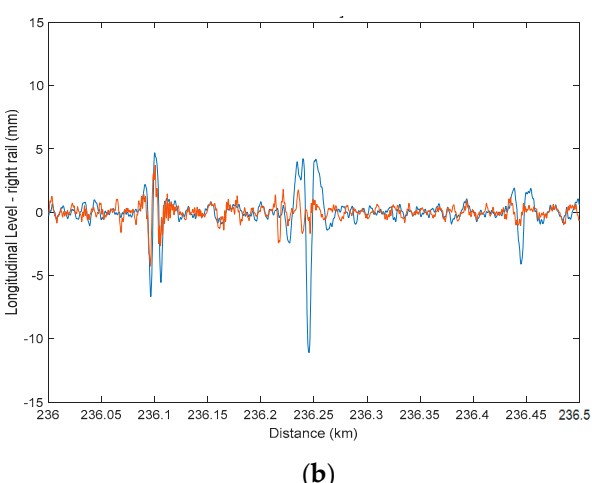

(**b**)

**Figure 8.** Synchronization of Longitudinal Level—right rail: (**a**) before and (**b**) after.

Although the track condition is only fully characterized by five indicators (longitudinal level, alignment, twist, gauge, and cross level), in this study, only the distributed irregularities in terms of longitudinal level and the alignment are considered because these are the two condition indicators whose degradation is more relevant for scheduling maintenance [13]. As presented in Figure 3, the distributed irregularities in terms of longitudinal level and alignment are characterized by the standard deviation of these two geometrical indicators over 200 m of track length. In this model application, a 51.4 km railway track subjected to 14 inspections actions with different time intervals is under analysis. Thus, a total number of 514 track segments (right and left rails) per inspection are under analysis. The dataset is unbalanced, as the interval between inspections (NDAYS) varies between 69 and 743 days, and the average time is approximately 183 days. For this reason, NDAYS is one of the variables considered in the study.

First, a descriptive statistical analysis of the standard deviations of the longitudinal level and the alignment of right and left rails was performed, including the computation of the main statistical measures (Table 1). The distributions of these variables, considered as potential predictors in logistic regression models, are illustrated by boxplots and histograms presented in Figure 9. In general, there are no substantial differences between the left and right rails. The distributions of these variables are moderately homogeneous as the coefficients of variation vary between 39% and 48%. However, the results reveal a positive asymmetry due to the existence of several higher outliers. SDLL takes values between 0.181 and 3.278 with a mean value equal to 0.763 on the left rail and 0.777 on the right rail. SDA takes values between 0.142 and 2.904 with a mean value equal to 0.632 on the left rail and 0.599 on the right rail.

For the geometrical track quality condition of the distributed irregularities characterized by the standard deviation of the longitudinal level (SDLL) and alignment (SDA) over 200 m segments, two conditions are defined as indicated in Table 2. These values are established according to [2].

**Table 1.** Descriptive statistical measures of the standard deviations of the longitudinal level and the alignment of right and left rails.

| Statistical Measures | $SDLL_L$ | $SDLL_R$ | $SDA_L$ | $SDA_R$ |
|---|---|---|---|---|
| Mean (mm) | 0.763 | 0.777 | 0.632 | 0.599 |
| Median (mm) | 0.687 | 0.699 | 0.575 | 0.544 |
| Standard Deviation (mm) | 0.370 | 0.372 | 0.261 | 0.238 |
| Coefficient of Variation | 48% | 48% | 41% | 39% |
| Skewness coefficient | 1.320 | 1.071 | 1.581 | 1.819 |
| Minimum (mm) | 0.211 | 0.181 | 0.142 | 0.157 |
| Maximum (mm) | 3.278 | 2.465 | 2.904 | 2.846 |
| Range (mm) | 3.068 | 2.284 | 2.763 | 2.689 |
| Inter-Quartile Range (mm) | 0.445 | 0.471 | 0.310 | 0.264 |

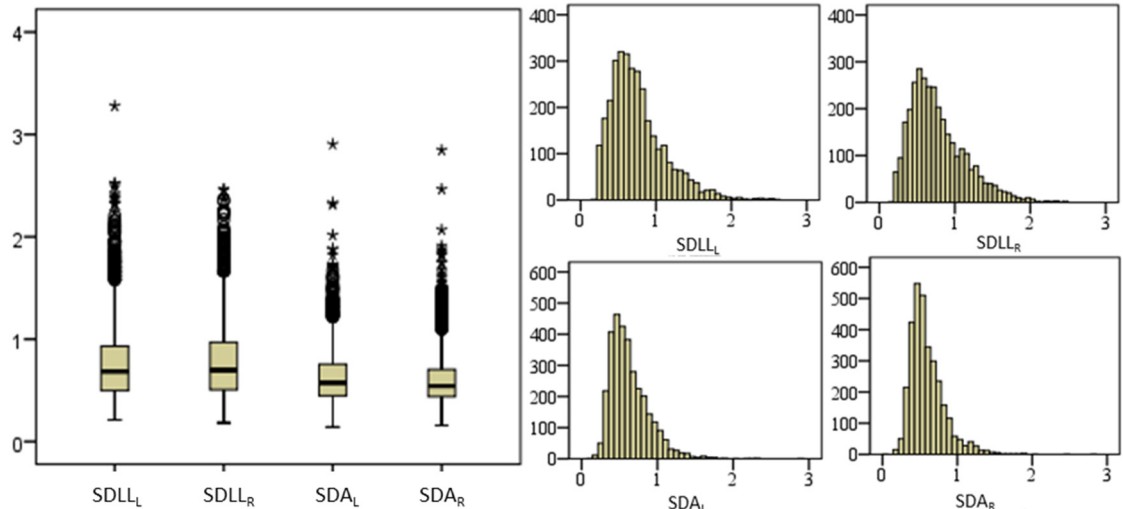

**Figure 9.** Boxplots and histograms of standard deviations of the longitudinal level and the alignment of right and left rails.

**Table 2.** Definition of geometrical track quality condition (*C*).

| Condition | SDLL (mm) | SDA (mm) |
|---|---|---|
| 0 | <1.2 | <0.8 |
| 1 | >1.2 | >0.8 |

*4.2. Selection of Independent Variables*

As previously mentioned, in the first phase of the modeling all independent variables have been considered as potential predictors of the logistic model, because all of them have a significant correlation with the dependent variable, that is, the geometrical track quality condition, *C*. This characteristic is shown by the correlation coefficients indicated in Table 3: all *p*-values are lower than 0.1% indicating statistical significance; the correlation coefficients are not null. There are strong correlations between the track condition (*C*) and SDLL and SDA, both higher than 0.5. From Table 3, the track condition has a greater relation to the track quality indicators of both rails than to the position of the track segment and the time interval between inspections. As expected, the track quality indicators highly depend on each other.

By taking into account the signal, it is still possible to check that the correlation is positive with SDLL and SDA, but negative with TSEG and NDAYS. Performing diagnosis of collinearity between independent variables, there is a strong correlation (matrix correlations in Table 3) between the variables related to longitudinal level and to the alignment, which is reinforced by higher VIF values (Table 3) than ideal ($\cong$1).

**Table 3.** Correlation coefficients and results of VIF and tolerance evaluation of the predictors.

| Variables | Correlation | | | | | VIF | Tolerance |
|---|---|---|---|---|---|---|---|
| | C | $SDLL_R$ | $SDLL_R$ | $SDA_L$ | $SDA_R$ | | |
| TSEG | −0.142 | −0.102 | −0.117 | −0.295 | −0.307 | 1.096 | 0.912 |
| NDAYS | −0.111 | 0.02 | 0.017 | −0.119 | −0.095 | 1.014 | 0.986 |
| $SDLL_L$ | 0.529 | | 0.889 | 0.464 | 0.556 | 4.548 | 0.22 |
| $SDLL_R$ | 0.526 | | | 0.468 | 0.546 | 4.383 | 0.228 |
| $SDA_L$ | 0.682 | | | | 0.769 | 2.158 | 0.463 |
| $SDA_R$ | 0.612 | | | | | 2.42 | 0.413 |

Given the importance of the SDLL and SDA variables to characterizing the geometrical track quality condition and the strong dependence between them, the next step of the modeling procedures was to perform a principal component analysis (PCA) with orthogonal (varimax) rotation of these four variables in order to make them independent variables. The application of the PCA technique resulted in two principal components, according to the eigenvalue criterion (greater than 1). A Kaiser–Meyer–Olkin measure verified the sample adequacy for analysis (KMO = 0.658). The Bartlett's sphericity test (chi-square = 7877.204, *p*-value < 0.001) also indicated that the correlations between the items were sufficient for analysis. These two principal components explained 90.04% of the total variance in the dataset. The high percentage of total explained variance and the values of Cronbach's Alpha being higher than 0.7 validated the obtained results (Table 4).

**Table 4.** PCA results.

| Dimension | Eigenvalue | Variance Accounted For | Cronbach Alpha |
|---|---|---|---|
| 1 | 1.878 | 46.945% | 0.761 |
| 2 | 1.724 | 43.093% | 0.735 |
| Total | | 90.038% | 1 |

The application of the principal component analysis resulted in two components. One, named LL, represented mostly the longitudinal level of both right and left rails ($SDLL_L$ and $SDL_R$) because the weights of $SDLL_L$ (0.565) and $SDL_R$ (0.590) are very high in this component. The other component, named AL, corresponded to the alignment of the two rails ($SDA_L$ and $SDA_R$). Even though the LL and AL components account for a high proportion of variance (90.038%), after the PCA analysis they were non-correlated variables, which allowed us to use them in the logistic regression to predict the track quality. As a result, these two principal components are represented as shown in Equations (5) and (6).

$$LL = 0.565\ SDLL^*_L + 0.590\ SDL^*_R - 0.222\ SDA^*_L - 0.108\ SDA^*_R \tag{5}$$

$$AL = -0.144\ SDLL^*_L - 0.180\ SDL^*_R + 0.652\ SDA^*_L + 0.560\ SDA^*_R \tag{6}$$

(*) standardized variables.

### 4.3. Construction of the Predictive Model

Following the methodology described in Section 3, a logistic equation was found to predict the geometrical track condition. The four variables resulting from the previous step, TSEG, NDAYS, LL, and AL, were considered predictors in the logistic regression modeling, and all of them satisfied the collinearity criterion, as VIF was near to 1 (Table 5).

The probability of occurrence of an event, namely, a track condition that requires maintenance, results from the logistic parameters presented in Table 6. In the model, the estimates α and β coefficients and the respective statistical significance are given by null *p*-values, which confirm that all of the coefficients are statistically significant, i.e., are significantly nonzero. After selecting the covariates, the fit of the model was tested and

the Hosmer–Lemeshow test was applied. The $r^2$ values according to Cox & Snell and Nagelkerke, as well as the percentage of the global success of the predictive model, are also presented in Table 6. The absolute value of β is a good indicator of the relevance of the parameter, highlighting that LL and AL are predictors with great impact on the track condition. All potential independent variables were considered in the logistic model. The statistical significance of the β coefficients given by the *p*-values implied inclusion in the model (Table 6). As indicated in bold values in Table 6, the probability of a railway track segment needing a maintenance action ($C = 1$) increases exponentially by 18.030 and 85.325 times with one unit of LL and AL, respectively. Considering the Hosmer–Lemeshow test, whose results are $Qui^2(8) = 6.110$ and $p = 0.635$, and having in mind the usual significance levels, the fitting does not reject the adequacy hypothesis from model to data. The coefficients for the evaluation indicate that the predictive model has good $r^2$ values (Table 6), meaning that the fitting with the logistic function is achieved.

**Table 5.** VIF and tolerance evaluation of the predictors.

| Variables | VIF | Tolerance |
|---|---|---|
| TSEG | 1.097 | 0.912 |
| NDAYS | 1.014 | 0.986 |
| LL | 1.006 | 0.994 |
| AL | 1.105 | 0.905 |

**Table 6.** Results of the logistic regression model.

| Variable | β | *p*-Value | Exp(β) | CI 95% | Cox & Snell $r^2$ | Nagelkerke $r^2$ |
|---|---|---|---|---|---|---|
| Constant | −2.126 | 0.000 | 0.119 | | | |
| TSEG | 0.007 | 0.000 | 1.007 | 1.005–1.009 | | |
| NDAYS | −0.001 | 0.001 | 0.999 | 0.998–0.999 | 0.575 | 0.803 |
| LL | 2.892 | 0.000 | 18.030 | 13.961–23.286 | | |
| AL | 4.446 | 0.000 | 85.325 | 58.394–124.679 | | |

To validate the model, the discriminatory power, sensitivity, specificity, and success rate are checked. The predictive model has a 91.1% success rate. Furthermore, the adjusted model has high sensitivity, i.e., correctly classifies 84.1% of inspections with $C = 1$, and a high specificity, i.e., correctly classifies 94.5% of inspections with $C = 0$. Moreover, the model has excellent discrimination ability as the area under the ROC curve is equal to 0.97 with *p*-value < 0.001.

*4.4. Validation*

To evaluate the performance, external validation of the model was performed. The logistic regression model was applied to predict the track condition at two inspection times (13th and 14th instant), corresponding to intervals of 203 and 395 (203 + 192) days, and the predicted results were validated with the real measurements. In Figure 10, a comparison between the predicted results (C_pred) and the observed measurements (C_obs) is shown for a track segment of 9 km, which corresponds to 45 track segments of 200 m. From this figure, at the 13th and 14th inspection instants, the track condition is correctly predicted in 37 out of 45 track segments, which is a good result.

Moreover, in these two inspection times, the value of the area under the ROC curve (0.907) and its confidence interval (0.879; 0.935) show that the model has excellent discriminatory power. The sensitivity, specificity, and success rate of the predicting model are 73.5%, 90.9%, and 85.3%, respectively. Given the hit rate, we can conclude that the model performs well in forecasting the geometrical track condition.

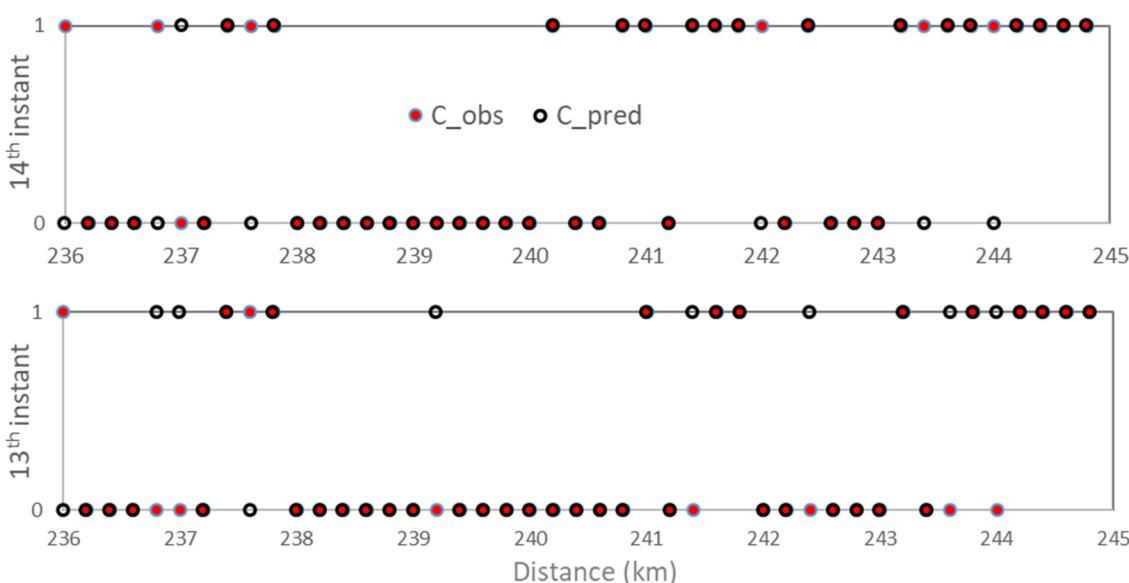

**Figure 10.** Model validation.

## 5. Conclusions

For scheduling maintenance activities of railway lines, the evolution of track degradation needs to be suitably predicted. Although several methodologies exist, the most accurate methodologies are complex to use and most of them do not consider the non-constant time interval of track inspections or the history of past track condition. To overcome these major limitations, a simple probabilistic model—a binary logistic function—was used to predict the geometrical track condition (dependent variable). The predictors (independent variables) are the standard deviation of the longitudinal level of the right and the left rail at each inspection time, the standard deviation of the alignment of the right and the left rail at each inspection time, the time interval between the inspection actions, and the sequential number of track segments for modeling. Given the strong dependence between the standard deviations of longitudinal level and alignment, a principal component analysis (PCA) was performed as a first phase of the modeling procedure. The quality of the fitting model was evaluated through $r^2$ coefficients of Cox & Snell and Nagelkerke, and validated with real measurements. The results prove that the model has a 91.1% success rate, an excellent discrimination ability, high sensitivity, classifying correctly 84.1% of inspections with need of maintenance, and high specificity, as it correctly classifies 94.5% of inspections with no demand of maintenance action.

**Author Contributions:** Conceptualization, C.V. and M.L.S.; methodology, C.V. and M.L.S.; validation, M.L.S.; writing, C.V. and M.L.S. All authors have read and agreed to the published version of the manuscript.

**Funding:** This work was financially supported by Base Funding—UIDB/04708/2020 and Programmatic Funding—UIDP/04708/2020 of CONSTRUCT—Instituto de I&D em Estruturas e Construções, funded by national funds through the FCT/MCTES (PIDDAC).

**Institutional Review Board Statement:** Not applicable.

**Informed Consent Statement:** Not applicable.

**Data Availability Statement:** Not applicable.

**Acknowledgments:** The data used in the research are confidential and have been kindly made available by Infraestruturas de Portugal.

**Conflicts of Interest:** The authors declare no conflict of interest.

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
