# Peer review of "Prediction of Railway Track Condition for Preventive Maintenance by Using a Data-Driven Approach"

_infrastructures, doi:10.3390/infrastructures7030034_

Round 1
Reviewer 1 Report
Dear Authors,
your manuscript is interesting, well written and organised and enjoyable to read. The methodology you present is very simple yet effective in the prediction of railway track conditions that makes it especially interesting for maintenance practitioners and infrastructure managers.
However, I would like to make some comments and suggestions that might improve the manuscript.
1. The figure captions are very short and sometimes not appropriate.
2. You make some statements about the PCA with respect to removing dependency and collinearity that are only true for normally distributed data. The independent variables you selected are not normally distributed as one can clearly see from figure 9. You should address/discuss that.
For more details please see the commented pdf attached.

Author Response
REPLY TO REVIEWER 1
Manuscript ID: infrastructures-1604316
Prediction of railway track condition for preventive maintenance by using a data-driven approach Authors: C. Vale and Simões M.L.
Date: 22/02/2022
We would like to start by expressing our appreciation to the Reviewer remarks addressed in the reports which helped us to improve the manuscript. This letter provides a point-by-point response to all the pertinent remarks of the Reviewer. The most relevant changes are identified in the manuscript by highlighted text.
- Specific Answers to the Reviewer’s Comments
Concerning the pertinent remarks raised by the Reviewer (italic text), the following changes were introduced into the manuscript, or alternatively the following clarification comments are made.
Dear Authors,
your manuscript is interesting, well written and organised and enjoyable to read. The methodology you present is very simple yet effective in the prediction of railway track conditions that makes it especially interesting for maintenance practitioners and infrastructure managers.
However, I would like to make some comments and suggestions that might improve the manuscript.
Comment 1. The figure captions are very short and sometimes not appropriate.
The captions of some of the Figures were rewritten more clearly in the revised manuscript.
Comment 2. You make some statements about the PCA with respect to removing dependency and collinearity that are only true for normally distributed data. The independent variables you selected are not normally distributed as one can clearly see from figure 9. You should address/discuss that.
Normal distribution is not necessary, because it is an exploratory approach. The Central Limit theorem empowers us to use the normal distribution even with no normal distribution if the sample is large enough, which is the case. Therefore, the non-normality of the data is not a problem. Even so, the data was standardized before PCA. For using PCA, the data may not be distributed normally but their linearity should exist as it is the case.
For more details please see the commented pdf attached.
Comment 3
Line 93 -Add reference if available.
A citation has been included in the manuscript, as suggested.
Comment 4
Figure 3 - The figure implies that the standard deviation (SD) of the longitudinal level and alignment is only taken from the wavelength range D1. But, this is in conflict with point i. and ii. below. This needs to be clarified.
Figure and description (points i. and ii. ) are both correct. D1 refers to the irregularity wavelength which is different from the track segment length where the standard deviation is calculated.
Comment 5
Figure 4 and 5 - Explain 'NL' and 'x'
The meaning of NL and x was included in the manuscript and also on the caption of Figures 4 and 5.
Comment 6
Lines 278 and 281 - The chosen independent variables are not normally distributed. This has an implication on the collinearity after applying the PCA which should be discussed; Strictly speaking, this is only true for normally distributed data.
An explanation has been done in Comment 2.
Reviewer 2 Report
1- In some of the figures, maybe it is better to explain more in the figures text and not just 3 words
2- Table 1 is divided in two pages, please correct it
Author Response
REPLY TO REVIEWER 2
Manuscript ID: infrastructures-1604316
Prediction of railway track condition for preventive maintenance by using a data-driven approach Authors: C. Vale and Simões M.L.
Date: 22/02/2022
We would like to start by expressing our appreciation to the Reviewer remarks addressed in the reports which helped us to improve the manuscript. This letter provides a point-by-point response to all the pertinent remarks of the Reviewer. The most relevant changes are identified in the manuscript by highlighted text.
- Specific Answers to the Reviewer’s Comments
Concerning the pertinent remarks raised by the Reviewer (italic text), the following changes were introduced into the manuscript, or alternatively the following clarification comments are made.
Comment 1- In some of the figures, maybe it is better to explain more in the figures text and not just 3 words
The captions of some of the Figures were rewritten more clearly in the revised manuscript.
Comment 2- Table 1 is divided in two pages, please correct it
The position of table 1 has been adjusted.
Reviewer 3 Report
The paper presents a logistic binary regression model to evaluate the degradation of track geometry.
The topic is interesting and the approach scientifically sound, however since the applied method is well-known in literature, the novelty of the application should be highlighted more in detail.
The Literature review should be improved including more recent studies and a comparison with previous data-driven approach should be included.
The following research could be mentioned:
- Pedro Cesar Lopes Gerum, Ayca Altay, Melike Baykal-Gürsoy, Data-driven predictive maintenance scheduling policies for railways, Transportation Research Part C: Emerging Technologies, Volume 107, 2019, Pages 137-154.
- Siddhartha Sharma, Yu Cui, Qing He, Reza Mohammadi, Zhiguo Li, Data-driven optimization of railway maintenance for track geometry, Transportation Research Part C: Emerging Technologies, Volume 90, 2018, Pages 34-58.
- Consilvio, A. Di Febbraro and N. Sacco, "A Rolling-Horizon Approach for Predictive Maintenance Planning to Reduce the Risk of Rail Service Disruptions," in IEEE Transactions on Reliability, vol. 70, no. 3, pp. 875-886, Sept. 2021, doi: 10.1109/TR.2020.3007504.
Author Response
REPLY TO REVIEWER 3
Manuscript ID: infrastructures-1604316
Prediction of railway track condition for preventive maintenance by using a data-driven approach Authors: C. Vale and Simões M.L.
Date: 22/02/2022
We would like to start by expressing our appreciation to the Reviewer remarks addressed in the reports which helped us to improve the manuscript. This letter provides a point-by-point response to all the pertinent remarks of the Reviewer. The most relevant changes are identified in the manuscript by highlighted text.
- Specific Answers to the Reviewer’s Comments
Concerning the pertinent remarks raised by the Reviewer (italic text), the following changes were introduced into the manuscript, or alternatively the following clarification comments are made.
Comment 1
The paper presents a logistic binary regression model to evaluate the degradation of track geometry.
The topic is interesting and the approach scientifically sound, however since the applied method is well-known in literature, the novelty of the application should be highlighted more in detail.
The novelty of the paper is mentioned in Line 60 and forward from Introduction and line 474 and forward from Conclusions. The proposed methodology is simple being able simultaneously to overcome some of the major limitations of the actual prediction models.
Comment 2
The Literature review should be improved including more recent studies and a comparison with previous data-driven approach should be included.
The following research could be mentioned:
- Pedro Cesar Lopes Gerum, Ayca Altay, Melike Baykal-Gürsoy, Data-driven predictive maintenance scheduling policies for railways, Transportation Research Part C: Emerging Technologies, Volume 107, 2019, Pages 137-154.
- Siddhartha Sharma, Yu Cui, Qing He, Reza Mohammadi, Zhiguo Li, Data-driven optimization of railway maintenance for track geometry, Transportation Research Part C: Emerging Technologies, Volume 90, 2018, Pages 34-58.
- Consilvio, A. Di Febbraro and N. Sacco, "A Rolling-Horizon Approach for Predictive Maintenance Planning to Reduce the Risk of Rail Service Disruptions," in IEEE Transactions on Reliability, vol. 70, no. 3, pp. 875-886, Sept. 2021, doi: 10.1109/TR.2020.3007504.
The authors thank to the reviewer for his comments. The first two references were included in the manuscript. The third reference is not included as the authors consider that the scope of Consilcio et al (2021) paper is not the same as the manuscript. In this paper, the authors propose a model for the risk-based scheduling of predictive maintenance activities on a railway line to intervene when a track segment has reached a certain state of degradation, while the present paper is focusing on the prediction track quality condition.